# Fractone Stem Cell Niche Components Provide Intuitive Clues in the Design of New Therapeutic Procedures/Biomatrices for Neural Repair

**DOI:** 10.3390/ijms23095148

**Published:** 2022-05-05

**Authors:** James Melrose

**Affiliations:** 1Raymond Purves Bone and Joint Research Laboratory, Kolling Institute, Northern Sydney Local Health District, St. Leonards, NSW 2065, Australia; james.melrose@sydney.edu.au; 2Graduate School of Biomedical Engineering, University of New South Wales, Sydney, NSW 2052, Australia; 3Sydney Medical School, Northern, The University of Sydney, Royal North Shore Hospital, St. Leonards, NSW 2065, Australia; 4Faculty of Medicine and Health, University of Sydney, Royal North Shore Hospital, St. Leonards, NSW 2065, Australia

**Keywords:** neural tissue repair, extracellular matrix, neural progenitor stem cells, perlecan, laminin, hyaluronan, stem cell niche

## Abstract

The aim of this study was to illustrate recent developments in neural repair utilizing hyaluronan as a carrier of olfactory bulb stem cells and in new bioscaffolds to promote neural repair. Hyaluronan interacts with brain hyalectan proteoglycans in protective structures around neurons in perineuronal nets, which also have roles in the synaptic plasticity and development of neuronal cognitive properties. Specialist stem cell niches termed fractones located in the sub-ventricular and sub-granular regions of the dentate gyrus of the hippocampus migrate to the olfactory bulb, which acts as a reserve of neuroprogenitor cells in the adult brain. The extracellular matrix associated with the fractone stem cell niche contains hyaluronan, perlecan and laminin α5, which regulate the quiescent recycling of stem cells and also provide a means of escaping to undergo the proliferation and differentiation to a pluripotent migratory progenitor cell type that can participate in repair processes in neural tissues. Significant improvement in the repair of spinal cord injury and brain trauma has been reported using this approach. FGF-2 sequestered by perlecan in the neuroprogenitor niche environment aids in these processes. Therapeutic procedures have been developed using olfactory ensheathing stem cells and hyaluronan as a carrier to promote neural repair processes. Now that recombinant perlecan domain I and domain V are available, strategies may also be expected in the near future using these to further promote neural repair strategies.

## 1. Introduction

Hyaluronan (HA) is a major space-filling component of the CNS/PNS providing tissue hydration and a matrix for cell attachment and an environment conducive to cellular survival and cellular migration during CNS/PNS development [1,2,3]. HA also ensures specific niche environments, and ionic gradients are maintained in the 3D brain architecture to ensure optimal conditions for cellular activity. The brain extracellular matrix (ECM) is unusual in that it is dominated by glycosaminoglycans (GAGs), particularly HA, and it is one of the softest tissues in the human body. The immobilization of HA in the CNS ECM is critical to the optimal functional properties of the brain; however, HA is a soluble polymer, and it relies on interactions with proteoglycans (PGs), HA receptors and HA interactive glycoproteins for its immobilization in the CNS/PNS ECM [2]. HA is a component of both diffuse and condensed brain ECM structures known as perineuronal nets (PNNs), which protect neurons and are essential for the maintenance of optimal neural cellular activity [4]. HA is the only non-sulfated GAG and has a relatively simple repeat N-acetyl-glucosamine-D-glucuronic acid disaccharide structure. HA is highly interactive with the lectican PGs and HA-receptors and also influences cell migration in developing tissues [5]. High molecular weight HA is anti-inflammatory and mops up free radicals generated by inflammatory cells; thus, it counters the development of neuro-inflammation. The formation of HA-lectican aggregates (particularly HA-aggrecan aggregates) is critical to tissue hydration, brain volume, maintenance of cellular organization and micro-compartmentalization in the brain ECM. This provides niche and metabolite gradient environments that promote optimal cellular activities in the brain 3D environment. The importance of HA’s roles in brain tissues becomes apparent in tissues that display an HA deficiency. For example, brain tissues that are deficient in HA synthase-3 activity (*Has3* KO) display frequent seizures and an epileptic phenotype [6,7].

## 2. Neuroprogenitor Stem Cell Niches and the Cell Regulatory Environment Provided by ECM Components

Schofield originally proposed the term stem cell niche in 1978 to define local hemopoietic tissue environments that had specific molecular and cellular activities that maintained hemapoietic stem cells [8]. Almost two decades earlier, Smart [9] and Altman and Das [10] had identified similar regions in the brain which they believed also provided an environment responsible for stem cell self-renewal or development into specific cell lineages that could migrate to sites of tissue development or areas undergoing ECM remodeling in tissue repair processes. In the brain, these regions were identified as the subventricular zone of the lateral ventricle and the sub-granular zone of the dentate gyrus of the hippocampus (Figure 1a). HA was subsequently identified as a key functional component of these brain stem cell niches controlling stem cell proliferation and differentiation [11]. Others have noted the importance of these niches in health and disease [12] and the important roles they play in tissue homeostasis [13,14] (Figure 1b).

Ependymal ciliated neuronal support cells (neuroglia) arising from embryonic neuro-ectodermal tissue in the epithelial lining of the ventricles of the brain and the central canal of the spinal cord have roles in the establishment of neural stem cell niches [11]. ECM and adhesion molecules maintain the ependymal niche architecture and provide interactive properties with stem cells that regulates the balance between stem cells that undergo quiescent recycling or a proportion of these cells that develop a migratory pluripotent neuroprogenitor phenotype [17]. Conservation of the cytoarchitecture of the ependymal niche is thus crucially important in the maintenance of stem cell viability and their differentiation into specific migratory cell lineages [18]. ECM molecules (laminins, perlecan) involved in neural stem cell–ependymal cell adhesion regulate interactions in the niche and actively participate in the regulation of neural stem cell dormancy/activation and the attainment of pluripotency [18]. The HS side chains of perlecan bind FGF-2, allowing this proteoglycan to act as a co-receptor, affecting the distribution of growth factors such as FGF-2 and chemokines in the niche environment that influence stem cell self-renewal and differentiation [19]. FGF-2 maintains stem cell viability and stimulates stem cell proliferation and differentiation, leading to the attainment of pluripotency.

In the adult mammalian brain, neural stem cells (NSCs) in the ventricular–sub-ventricular zone contain a subpopulation of cells with astroglial properties (B1 cells), which give rise to intermediate transient amplifying progenitors (IPCs, C cells), which transit to the olfactory bulb as a stem cell reservoir. NSCs and IPCs are the primary and secondary progenitor cells of the niche. These cells have long processes that reach into distant regions of the niche, including the adjacent vasculature. NSCs are thus in a unique cellular compartment, and depending on ECM components, they are exposed and may be stimulated to proliferate and undergo differentiation or they may remain as quiescent cells which continue to slowly recycle for self-renewal. Cell–cell, cell–ECM and soluble factor interactions involving HA, perlecan, laminin and FGF-2 thus all have roles in the regulation of NSC behavior. To better understand the complexity of the niche compartment, it has been proposed that it should be considered as three functional regions, proximal (zone I), intermediate (zone II) and distal (basal zone III) [20]. These regions are shown schematically in Figure 2. NSCs corresponding to type B1 cells are shown in blue, these are bordered by multiciliated ependymal cells (E) lining the ventricle. The B1 cells give rise to IPCs or C cells indicated in green. These are the transit-amplifying cells that generate the neuroblast type A cells indicated in red. The B1 cells have an epithelial morphology and contain a thin apical process and primary cilium that contacts the lateral ventricle (V), and a long basal process that extends as a foot-like process that contacts adjacent blood vessels (BV) labeled purple. A schematic depiction of the SVZ niche organization is depicted in Figure 2. 

The specialized neural stem cell niche has been termed the fractone in honor of the late mathematician Dr Benoit Mandelbrot due to their fractal-like appearances similar to Dr Mandelbrot’s early computer graphic generated fractal geometric images [21]. The identification of fractones in the spinal cord [22] suggested that resident cord stem cells might be useful in the promotion of recovery from spinal cord injury [23]. Genetic fate mapping showed that almost all of the spinal cord neural stem cell potential resided within the ependymal cells that line the central canal of the spinal cord [22]. These cells are recruited following spinal cord injury to produce not only the scar-forming glial cells, but also, to a lesser extent, the oligodendrocytes that remyelinate damaged axons [24]. The possibility therefore exists that ependymal-generated neuroprogenitor cells could also be harnessed for the endogenous repair of neural tissues in the CNS/PNS guided by key regulatory functional components of the niche environment [25,26,27].

Fractone ECM structures in the neural stem cell niche [17] influence neural stem and progenitor cell formation, proliferation, and/or maintenance [16,28,29,30,31,32,33] (Figure 1b). In repair biology in brain tissues, the aim is to mimic this niche environment experimentally to control neuroprogenitor cell activity in vivo [34,35]. Ependymal cells are the source of laminin α5-containing fractone bulbs [33,36]. Deletion of laminin α5 from ependymal cells results in a 60% increase in niche cell proliferation, indicating that laminin α5 modulates the proliferative status of the neural stem cell niche. The C-terminus of the five laminin α chains are key to laminin signaling and are crucial for pluripotent stem cell survival and self-renewal bin vitro [32,37,38] and inhibit stem cell proliferation in vivo [39]. Perlecan interacts with laminin in the niche environment and with FGF-2 to promote niche cell proliferation and with BMP-4 and BMP-7 to inhibit niche cell proliferation [32,37,38,40].

A reduction in HS 6-O-sulfation, which is critical for FGF-2 signal transduction, has been observed in the aged sub-ventricular zone, and this reduces cell proliferation through a failure of FGF-2-induced phosphorylation of extracellular signal-regulated kinase (Erk1/2) [41,42]. Increases in HS 6-O-endo-sulfatase has also been observed in the aged sub-ventricular zone responsible for observed modifications in HS sulfation [41,42]. An appropriately assembled fractone ECM is important for correct brain function. A loss of perlecan or laminin from the fractone ECM results in impaired brain functional properties observed in autism in humans and is also observed in BTBR T+ tf/J mice, which is an animal model of autism [43,44]. HA and perlecan are components of stem cell niches in the intervertebral disc [45], human fetal cartilaginous rudiments of the hip and knee joints [38], and fetal human elbow [46]; thus, it is not surprising that they are also functional components of fractone stem cell niche structures [47].

## 3. Development of HA Hydrogel Cell Delivery and Therapeutic Biomatrices for Tissue Repair

HA is the most abundant GAG of neural tissues and, as already discussed in the introduction, has important space filling and hydrating properties and is a cell-friendly ECM component that promotes cell viability and proliferative processes important in tissue development and repair processes. HA is thus a logical candidate to investigate for potential application in neural repair biology. In its native form, HA is a weak scaffolding material, and for it to remain in tissues, it must interact with HA receptors and HA interactive PGs and glycoproteins; the large size of these aggregates physically entraps these HA complexes within collagenous networks in tissues. However, neural tissues lack these collagenous networks, and the immobilization of HA in neural tissues relies on interaction with lectican PGs and neural HA receptors. This is essential to provide HA’s longevity in tissues, since it is rapidly degraded in vivo by hyaluronidase and is highly soluble, with the generated HA oligosaccharides rapidly undergoing dispersal from tissues. For tissue engineering applications HA must therefore be chemically modified and crosslinked or attached to another polymer to form stable, functional composite scaffolds that support cell adhesion and proliferation [48,49,50]. HA is a versatile scaffolding material amenable to crosslinking using a number of chemical methods under basic, acidic, and neutral pH conditions or to the production of composites with other natural and synthetic polymers to confer strength [51]. This facilitates the use of HA in diverse applications [35,52,53,54,55] to improve the healing of wounds, burns, and traumatized tissues that require a space-filling scaffold that preserves tissue hydration providing a matrix conducive to the attraction of cell populations into defect sites to affect tissue repair processes [45,50,56]. HA’s interactions with neural CS lectican PGs form dense ECM surrounding neurons termed perineuronal nets (PNNs) (Figure 3a). These have neuroprotective properties, convey neural and synaptic plasticity and have roles in memory and cognitive learning. Monoclonal antibody 1-B-6 identifies reducing terminal stub epitopes in the lectican PGs following chondroitinase ABC digestion and can be used to immunolocalize PNN structures in brain tissues (Figure 3b).

A diverse range of HA bioscaffold applications have been developed for tissue repair (Table 1). HA is a useful multifunctional biomaterial that has been employed in repair strategies on infarcted myocardial tissues [57],stem cell delivery for corneal repair [58], vocal cord [59], endometrial [60] endodontic tissue repair [61]. HA has been used as a stem cell delivery vehicle for neural crest cells [62] and improves tissue regeneration. Biomimetic HA hydrogels have also found application as wound healing agents [63]. A diverse range of HA-based strategies have also been developed for the repair of neural tissues. Biocompatible injectable methacrylated gellan gum HA biocomposites functionalized with manganese have been developed that can be used to administer human-derived adipose stem cells using a T1-weighted MRI image-guidance system for stem cell delivery to specific brain regions for the treatment of amyotrophic lateral sclerosis. This HA hydrogel ensured that the viability of the injected stem cells was maintained at the injection site for at least 14 days [64].

HA can be modified using a number of crosslinking chemistries to alter its viscoelastic properties to prepare scaffolds that mimic the material properties of native brain tissue, which is one of the softest tissues in the human body [65]. Divinyl sulfone crosslinked HA has been used to prepare scaffolds with varying pore sizes that support cell migration and neurite extension HA–biocomposites have also been prepared that are suitable for brain repair applications [66]. HA hydrogels suitable for gene delivery have been used in brain repair strategies; these hydrogels provide a macroporous structure to the injection site to promote cell migration and proliferation and repair processes [67]. An HA film sheath has been used for VEGF gene therapy to treat peripheral nerve damage [68]. Peptides such as laminin IKVAV and RGD have also been attached to HA to provide hydrogels and biocomposites with improved cell attachment and tissue repair properties [69,70,71]. Laminin-IKVAV peptide improves neural cell attachment, proliferation and neurite extension [69,70,71]. IKVAV-HA implants inserted into brain defects were invaded by blood vessels, glial cells and axons to promote new tissue formation by 6 weeks of implantation. Integrin binding RGD peptide attached to hydroxyphenyl-modified HA has also improved cell-binding properties and has been applied in spinal cord repair [72].

Biomimetic composites of HA with collagen, laminin and CS-PG mimic the native brain structure to promote cell survival and neural differentiation and neural outgrowth in brain repair strategies [63,73]. HA–laminin hydrogels implanted into brain defects inhibited glial scar formation at the defect site and promoted neural proliferation and neurite extension in the repair site [74]. A 3D gelatin–HA scaffold has been applied to the repair of spinal cord defects. This scaffold undergoes in situ gelation to fit the cord defect perfectly through visible light induced crosslinking [75]. The HA component of this scaffold reduces inflammation, inhibits glial scar formation, promotes endogenous NSC migration and neurogenesis, neuron maturation and axonal regeneration in the defect site. Complete spinal cord repair has been reported using this bioscaffold. Biocomposite scaffolds of recombinant spider silk protein (spidroin) and HA have also been prepared and applied to spinal cord repair [76]. Biocomposites made from electrospun fibers of HA–polycaprolactone provide high-porosity nanofibrous scaffolds suitable for the growth of SH-SY5Y human neuroblastoma cells. HA–collagen 3D composites containing electrospun polycaprolactone fibers have also yielded promising results in the regeneration of peripheral nerves [77]. HA-poly-D-lysine copolymer hydrogel has an open porous structure and viscoelastic properties similar to those of native brain tissue [78]. This hydrogel has also been proposed to be a suitable scaffolding material for peripheral nerve regeneration. Biocomposites of HA containing PLGA microspheres loaded with VEGF and BDNF promote neural growth, and the extended release of these factors promotes neural repair over an extended time frame [79]. A neurotrophin NGF–HA hydrogel filler has also been applied to the repair of a 10 mm sciatic nerve defect [80].

### Application of HA as a Delivery Vehicle for Olfactory Ensheathing Cells ± Mesenchymal Stem Cells from a Number of Tissues for Neural Repair

Olfactory ensheathing cells (OECs) support axonal regeneration and remyelination with the appropriate formation of axonal nodes of Ranvier and improvement of nerve conduction velocity. OECs derived from nasal mucosa are of clinical interest, since these cells are amenable for harvest for such autotransplantation procedures [85]. Research conducted over the last decade has shown that neuroprogenitor stem cells harvested from the olfactory bulb have considerable potential in the promotion of neurogenesis and neural repair processes [86,87,88,89]. Adult OECs display considerable potential in the regeneration of traumatically injured neural tissues [90] and in the repair of spinal cord injuries [86,91,92]. The tissue reparative potential of bone marrow-derived mesenchymal stem cells and OECs have been compared in the repair of the injured rat spinal cord [93]. NSCs and OECs display synergistic effects in the repair of adult spinal cord injuries [94] and traumatic brain injuries in rats [95], including transient focal cerebral ischemia [96]. Olfactory ensheathing neuroprogenitor stem cells and human umbilical cord mesenchymal stem cell-derived exosomes have also been shown to promote sciatic nerve regeneration [97]. Adipose-derived stem cells and olfactory ensheathing neuroprogenitor stem cells have also been used in combination therapy to treat spinal cord injured rats [98]. OECs have been used in multi-layered conductive nanofibrous conduit scaffolds in the repair of peripheral nerve damage in rats [99]. The development of electroconductive bioscaffold delivery systems for neural stem cells and spinal cord repair is an exciting development given that neural cells are one of the most sensitive cells to electrostimulation in the human body [100].

## 4. Application of HS Containing Biomatrices for Neural Repair

Perlecan is also a major functional component of the stem cell niche and has many attributes with regard to tissue development and repair processes [101,102,103,104]. Perlecan is expressed in the basal neuroepithelium during neural development and is a crucial component of the neural niche [39,105]. Perlecan has multifunctional instructive properties [106] in developmental brain tissues [103,107] and promotes the proliferation and differentiation of neuroprogenitor stem cells in the sub-ventricular fractones through the sequestration of FGF-2 in the neural niche activating the Akt and Erk 1/2 cell signaling pathways [37]. The Wnt and ShH pathways also regulate stem cell proliferation, neurogenesis and neural network formation [108,109]. However, Wnt and Hedgehog proteins are relatively poorly soluble in aqueous media. Wnt and ShH bind to perlecan domain II, and this acts as a transport PG, aiding in the establishment of Wnt and ShH morphogen gradients in tissues that are important for tissue development [101]. The availability of recombinant perlecan domain I and domain V will allow investigations to be undertaken in the stimulation of neural repair processes in tissues [110,111,112,113] and in repair of the blood–brain barrier following ischemic stroke [114,115,116]. Further studies with perlecan in neural repair processes are expected in the future, and these offer exciting possibilities. 

### 4.1. Harnessing Cell Instructive Properties of Perlecan’s HS Side Chains in Repair Biology

The fine structure (sulfation position and density) of the HS side chains of perlecan is an important regulatory determinant in the differentiation of pluripotent stem cells in the niche environment in neural tissues [117]. Interaction of HS with growth factors (FGF-2) and morphogens (Wnt, SHh) is also essential for the long-term viability of recycling stem cells and the proliferation and differentiation of stem cells that have escaped from quiescent recycling and along with interactions with niche ECM components regulates the development of stem cell lineages that attain migratory properties facilitating their participation in neural repair processes [40,104,114]. The expressions of HS biosynthetic enzymes in the niche and tissue environments also have important roles in determining the fine structure of HS and how it exerts these effects spatially and temporally in tissue development and neural repair processes and also has roles in the determination of synaptic specificity, axonal guidance, synapse development and synapse function [118]. Perlecan is an important regulatory cell instructive PG in the neural stem cell fractone [119]. The availability of recombinant perlecan domain I and domain V now makes it possible to incorporate these components into new generation bioscaffolds in neural repair strategies attempting to mimic the niche environment of native neural tissues. Such approaches used in combination with HA and neural progenitor stem cell preparations have a high probability of further improving on existing neural repair applications.

Collagen–HS porous scaffolds containing NSCs have been used to treat a rat model of traumatic brain injury, established using a controlled cortical impact [120]. Brain edema and cell apoptosis were significantly reduced, and motor and cognitive functions markedly improved using this procedure suggesting that porous collagen–HS scaffolds loaded with NSCs can improve neurological deficits in a rat model of traumatic brain injury [120]. Three-dimensional (3D) bioprinter-assembled collagen–HS scaffolds have also been used to treat controlled spinal cord injuries in rats [121]. The HS component of this scaffolding material crosslinks the collagen fibers, increasing its compression modulus and mechanical stability. This scaffold displays good biocompatibility with neurons co-administered within the scaffold. The HS component of this scaffold significantly improves the immobilization of bioavailable FGF-2, which promotes progenitor cell proliferation. A significant recovery in locomotor function and increased numbers of neurofilament positive cells were evident using this approach, suggesting that this matrix actively stimulates axonal guidance and neural repair processes. Porous bioscaffolds of chitosan–gelatin containing HA and/or HS have also been used in neural tissue engineering [122]. Such scaffolds contained highly interconnected pores ranging in size from 90 to 140 μm, and the scaffold had a porosity index of over 96%. Neural progenitor stem cells seeded into this matrix displayed adhesion, proliferation and multi-lineage differentiation in the 3D scaffold environment, indicating that this matrix may be useful in neural repair biology applications [122].

### 4.2. Development of Artificial Neural Stem Cell Niches

Significant improvements in bioscaffold microfabrication methodology has permitted the miniaturization of these platforms. Lithography and direct laser printing have been applied to prepare 2D patterns and 3D scaffolds to shape hydrogels and synthetic polymers to create niche-like structures for single neural cell culture [123]. Artificial laminin 3D neural stem cell niche-like structures have been developed to recapitulate the dynamic nature and some of the biological complexity of the neural stem cell niche and maintain laminin in a native conformation and orientation as found in the niche. These scaffolds support enhanced human NSC proliferation and neurite extension [124,125]. Stem cell niches are intricate spaces that provide specific chemical and biological environments that control stem cell fate [126]. Microdevices have been developed that have proved useful for the culture of NG108-15 neuroblastoma and human NPCs and represent a system amenable to modifications that promote these cellular activities for applications in neural repair biology [121,122,124].

## 5. Conclusions

With a greater understanding of the roles of key functional components in the stem cell niche environment that promote neurogenesis, repair and regeneration following trauma, it is logical that these components should be developed for use in neural repair strategies. The use of neural stem cells from the olfactory bulb and HA either as a stem cell delivery vehicle or as a component of composite new generation bioscaffolds is a powerful synergistic combination that is enabling the more effective repair of neural tissues. By understanding the roles of HA in the fractone stem cell niches of the sub-ventricular zone and sub-granular region of the dentate gyrus of the hippocampus, it has been possible to formulate tissue repair procedures using HA either as a delivery vehicle or as a component of composite bioscaffolds to improve neural repair. Further components of the niche environment such as perlecan and laminin also regulate stem cell proliferation and differentiation, and these are promising agents that may also find application in novel neural repair strategies in the future. Perlecan has already shown promise in the repair of the blood–brain barrier following ischemic stroke.

## Figures and Tables

**Figure 1 ijms-23-05148-f001:**
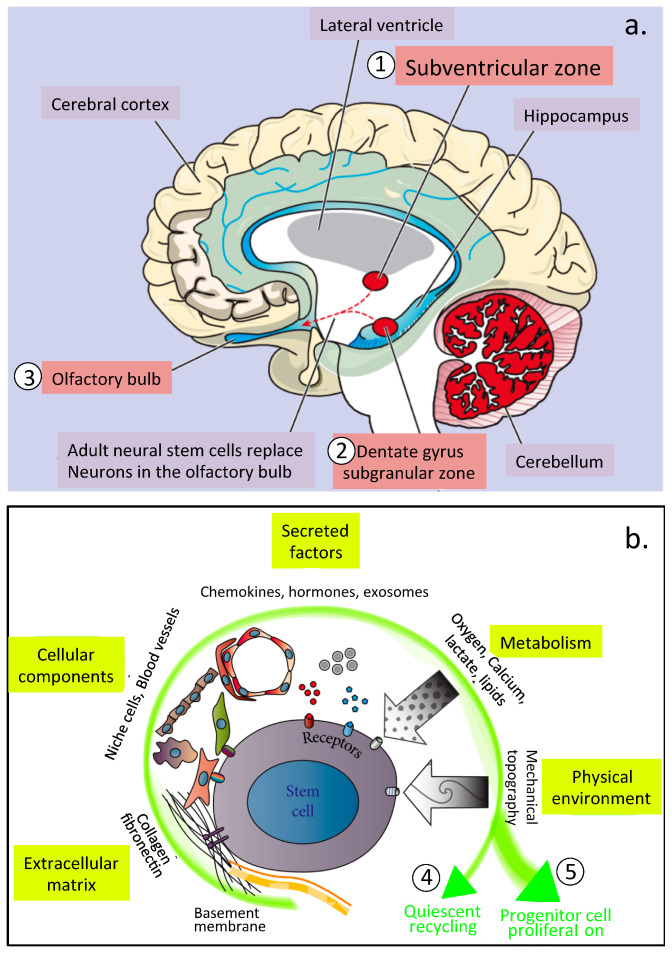
Schematic depiction of a cross-section through the human brain showing the two main regions of neuroprogenitor cells in the ventricular (1) and sub-granular regions of the dentate gyrus of the hippocampus (2) and the olfactory bulb (3), a storage region for neuroprogenitor cells (**a**). Segment (**b**) depicts schematically the multiple intrinsic and extrinsic influences exerted on stem cells in the niche micro-environment that determine whether stem cells undergo quiescent recycling (4) or attain a pluripotent migratory stem cell phenotype and escape from the regulatory niche (5) to participate in tissue development or tissue repair. Segment (**a**) reproduced from [15] with permission, © MA Healthcare Ltd 2008. Segment (**b**) reproduced from [16] with permission. Copyright © 2018 Sari Pennings et al. reproduced under the Creative Commons Attribution License.

**Figure 2 ijms-23-05148-f002:**
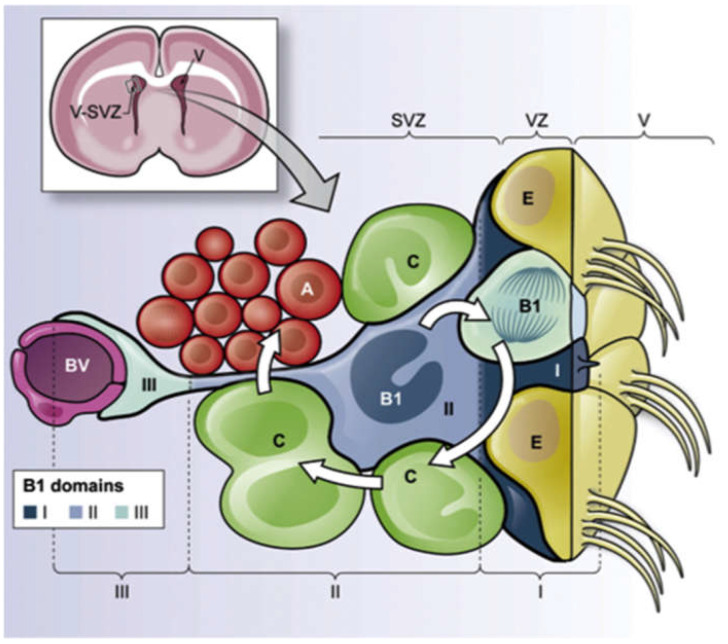
Schematic depiction of the SVZ stem cell niche location in a frontal cross-section of adult brain (upper figure) and the organization of the fractone niche showing regions where laminin, perlecan, FGF-2 and HA interact with the niche cell populations. Functional domains of the B1 primary stem cell niche cells are indicated. Domain I (proximal or apical, dark blue) contains the primary cilium and is in direct contact with the CSF and soluble factors and signaling molecules from neighboring ependymal cells. Domain II (intermediate, medium blue) is where IPCs, neuroblasts, neuronal terminals and cell–cell interactions occur between B1 cells and their progeny. In Domain III (distal, light blue), the B1 cell contains a specialized end-foot process that contacts BVs where blood-borne and endothelial cell generated factors may stimulate the B1 cells. The different niche cell populations are labeled as neuroblast type A cells (red), IPCs type C cells (green), B1 cells (light and dark blue), and ependymal cells (brown). The ventricle (V), ventricular zone (VZ) and sub-ventricular (SV) regions of the fractone are indicated. Figure reproduced with permission from [20], © Penning et al 2018.

**Figure 3 ijms-23-05148-f003:**
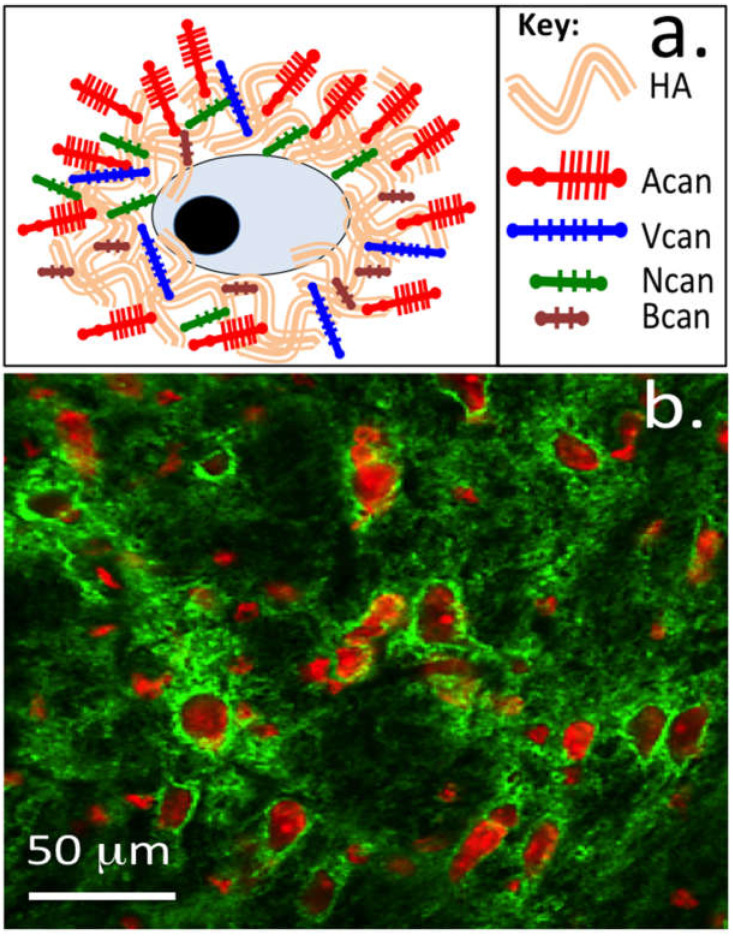
Schematic of aperineuronal net structure showing the interaction of hyaluronan (HA) with members of the lectican proteoglycan family, aggrecan (Acan), versican (Vcan), Brevican (Bcan) and neurocan (Ncan) showing the dense matrix around neurons termed perineural nets (PNNs) (**a**). The 1-B-6 (+) proteoglycan immunolocalizations shown in (**b**) depicting perineuronal nets are modified with permission from [47,56], © Caterson and Hayes 2002.

**Table 1 ijms-23-05148-t001:** The Versatility of HA in the Development of Hydrogels and Bioscaffolds for Tissue Repair.

HA Hydrogel/Scaffold and Its Properties in Tissue Repair Processes		Ref
Injectable HA hydrogel	MSC repair of infarcted myocardium.	[57]
Tissue adhesive HA hydrogel	Sutureless stem cell delivery and regeneration of corneal epithelium and stroma.	[58]
HA hydrogel	MSC delivery to damaged vocal cord.	[81]
HA hydrogel	Treatment of Endometrial Injury in a Rat Model of Asherman’s Syndrome.	[60]
Injectable HA hydrogel	Tunable HA hydrogels releasing chemotactic and angiogenic growth factors for endodontic regeneration.	[61]
HA scaffold	Scaffolds that improve stem cell functions for tissue repair and regeneration.	[82]
Interpenetrating collagen, HA, polymer networks	Scaffolds for brain tissue engineering.	[53]
Injectable HA Scaffolds with Macroporous Architecture	Scaffold designed for gene delivery for tissue repair.	[67]
Combination of hyaluronic acid hydrogel scaffold and PLGA microspheres	Extended delivery of VEGF and BDNF from PGLA microspheres promotes neural growth.	[79]
Divynyl sulfone crosslinked HA	Scaffolds with a range of pore sizes supporting cell migration and neurite extension.	[83]
Neurotrophin NGF-HA hydrogel filler cell delivery system	Scaffold filler hydrogel used in combination with olfactory ensheathing cells to repair of a 10 mm gap model of sciatic nerve injury in Sprague–Dawley rats	[80]
Biomimetic collagen, laminin, HA, and CS–proteoglycan biocomposites	Biomimetic hydrogels of collagen, laminin, HA, and CS-PGs developed to reproduce native ECM structure for the promotion of cell survival, neural differentiation, and neurite outgrowth.	[73]
Electrospun HA–polycaprolactone nanofiber bioscaffolds	Electrospun high-porosity nanofibrous scaffolds suitable for the growth of SH-SY5Y human neuroblastoma cells.	[84]
HA-poly-D-lysine hydrogel	Copolymer hydrogel with an open porous structure and viscoelastic properties similar to those of native brain tissue. Proposed as a promising scaffold for the repair of brain defects.	[78]
HA–laminin hydrogels	HA–laminin hydrogels implanted into brain defects promoted neurite extension and inhibited glial scar formation.	[74]

## Data Availability

All data is contained within each cited study.

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
