# Peer review of "Fractone Stem Cell Niche Components Provide Intuitive Clues in the Design of New Therapeutic Procedures/Biomatrices for Neural Repair"

_ijms, 2022, doi:10.3390/ijms23095148_

Round 1
Reviewer 1 Report
The authors describe about the important instructive contributions of hyaluronan in neural structures, neural cell regulation and its potential application in neural repair biology. Although this manuscript is well-written, I would like to point out several points to be improved.
- There are some overlap between this manuscript and the published article from the same author (Int. J. Mol. Sci. 2021, 22, 5583. https://doi.org/10.3390/ijms22115583) . It's better to decrease the overlap by revising the manuscript. (Otherwise, this manuscript won't be accepted.)
- The organization of abstract is not good. The author should revise the abstract to follow the standardized manner.
- Figure 1 is not high quality. The author should draw the illustration more clearly and beautifully and revise the figure legend so that the readers can understand the schemes visually.
Author Response
Reviewer 1 COMMENTS
The authors describe about the important instructive contributions of hyaluronan in neural structures, neural cell regulation and its potential application in neural repair biology. Although this manuscript is well-written, I would like to point out several points to be improved.
- There are some overlap between this manuscript and the published article from the same author (Int. J. Mol. Sci. 2021, 22, 5583. https://doi.org/10.3390/ijms22115583) . It's better to decrease the overlap by revising the manuscript. (Otherwise, this manuscript won't be accepted.)
- The organization of abstract is not good. The author should revise the abstract to follow the standardized manner.
- Figure 1 is not high quality. The author should draw the illustration more clearly and beautifully and revise the figure legend so that the readers can understand the schemes visually.
Author responses
The manuscript has been revised removing content overlap with Int. J. Mol. Sci. 2021, 22, 5583. https://doi.org/10.3390/ijms22115583 and re-organised with additional comments on HA as a scaffolding material, all new material is highlighted in the marked up revised manuscript. A revised figure 1 is provided and a new figure 2 to explain fractone organisation better. Table 1 has also been modified with the inclusion of additional information.
Reviewer 2 Report
Dear author
in my opinion the review entitled "The Niche Components of Fractone Stem Cells Provide Intuitive Clues in the Design of Novel Therapeutic / Biomatic Procedures for Neural Repair"
does not seem entirely and completely clear. Even though the very interesting and innovative argument in the field,
it does not seem to have original ideas with regards to published literature and could be presented in a more structured-way. Conversely references are appropriate.
Also, the figure should be more explicative. Table 2 seems inappropriate. in coclusion the review is difficult to read and interpret.
Author Response
Reviewer 2 COMMENTS
Dear author
in my opinion the review entitled "The Niche Components of Fractone Stem Cells Provide Intuitive Clues in the Design of Novel Therapeutic / Biomatic Procedures for Neural Repair" does not seem entirely and completely clear. Even though the very interesting and innovative argument in the field, it does not seem to have original ideas with regards to published literature and could be presented in a more structured-way. Conversely references are appropriate.
Also, the figure should be more explicative. Table 2 seems inappropriate. in coclusion the review is difficult to read and interpret.
Author responses
The manuscript has been extensively revised with additional comments on HA as a scaffolding material, all new material is highlighted in the marked up revised manuscript. A revised figure 1 is provided and a new figure 2 has been added to better explain fractone organisation and interactions between cell populations in the niche environment. Table 2 has been deleted and a new more informative Table 1 provided.
Round 2
Reviewer 1 Report
The author revised the manuscript adequately.
Reviewer 2 Report
Dear author
I appreciated the article and the corrections made. Moreover, I wonder if there are any trials in which HA has been used for neurodegenerative diseases. The potential applications of this material should be extensively investigated.